# Ultra-compact exciton polariton modulator based on van der Waals semiconductors

Seong Won Lee[1,2,3], Jong Seok Lee[1,2,3], Woo Hun Choi[1,2], Daegwang Choi[1,2] & Su-Hyun Gong [1,2] ✉

With the rapid emergence of artificial intelligence (AI) technology and the exponential growth in data generation, there is an increasing demand for high-performance and highly integratable optical modulators. In this work, we present an ultra-compact exciton-polariton Mach–Zehnder (MZ) modulator based on $WS_2$ multilayers. The guided exciton-polariton modes arise in an ultrathin $WS_2$ waveguide due to the strong excitonic resonance. By locally exciting excitons using a modulation laser in one arm of the MZ modulator, we induce changes in the effective refractive index of the polariton mode, resulting in modulation of transmitted intensity. Remarkably, we achieve a maximum modulation of −6.20 dB with an ultra-short modulation length of 2 μm. Our MZ modulator boasts an ultra-compact footprint area of ~30 μm² and a thin thickness of 18 nm. Our findings present new opportunities for the advancement of highly integrated and efficient photonic devices utilizing van der Waals materials.

The rapid rise of artificial intelligence (AI) technology and the exponential surge in data generation have created a pressing need for high-performance and highly integratable optical modulators[1,2]. These modulators are essential elements in photonic integrated circuits, enabling the manipulation of light signals for data transmission, processing, and communication. Traditional light modulation techniques, such as electro-optic[3–9], acoustic-optic[10–13], and magneto-optic methods[14,15], rely on altering the effective refractive index of optical-guided modes. And, the miniaturization of optical modulators plays a crucial role in enhancing the integration level of photonic integrated circuits. However, conventional modulation approaches often exhibit limited changes in refractive index, requiring long modulation lengths and resulting in large device footprints. Therefore, achieving substantial light modulation with reduced device size is a significant challenge in the field of optical modulators.

To address this challenge, various strategies have been pursued. One approach involves integrating light-modulating components with other materials such as electro-optic polymers[16–19] or 2D materials[8,20–22], enabling enhanced refractive index variation. Another avenue involves reducing the mode volume of guided light by utilizing photonic crystals[16,23–25] or plasmonic structures[5,9,26–31], thereby promoting enhanced light-matter interactions. Notably, hybrid structures incorporating plasmonic modes have demonstrated remarkable achievements with a modulation length of a few μm[5,30,31]. Additionally, the incorporation of a micro-ring resonator or Fabry–Perot cavity can reduce the modulator's footprint by enabling multiple round-trip interactions of light within a closed loop[3–5,32–34]. The use of a ring resonator, however, exhibits limited spectral bandwidth due to the narrow linewidth of resonance frequencies. Despite those extensive efforts, the footprint of current light modulators remains typically larger than ~100 μm² [35].

Exciton polaritons, resulting from the strong interaction between excitons and photons, offer a promising platform for efficient modulators owing to their highly interactive characteristics[36–39]. Their strong repulsive interaction can be harnessed to generate a local potential gradient, enabling effective modulation of the polariton phase and amplitude. However, the realization of such exciton polariton modulator devices necessitates strict conditions, notably achieving a strong coupling regime. This regime demands sophisticated fabrication processes to attain high-quality distributed Bragg reflectors, often resulting in a relatively thick system (>5 μm) and requiring extended growth times. Furthermore, it is important to note that most

¹Department of Physics, Korea University, Seoul 02841, South Korea. ²KU Photonics Center, Korea University, Seoul 02841, South Korea. ³These authors contributed equally: Seong Won Lee, Jong Seok Lee. ✉e-mail: shgong@korea.ac.kr

of the demonstrated polariton modulators have relied on exciton-polariton condensation to achieve the necessary potential gradient for polaritons, a task that poses even greater challenges.

In this study, we present an ultra-compact exciton-polariton modulator utilizing $WS_2$ multilayers. The guided exciton polariton naturally emerges in the near-field of a bare $WS_2$ layer due to the strong coupling between excitons and photons. By controlling the exciton energy through non-resonant optical pumping, we achieve easy tunability of the effective refractive index of the polariton mode. To demonstrate polariton modulation, we implement local non-resonant pumping in one arm of a $WS_2$ Mach–Zehnder (MZ) interferometer. Despite a short modulation length of 2 μm, we successfully achieved a remarkable modulation ratio of −6.20 dB. Our MZ modulator features an ultra-compact footprint area of ~30 μm² and a thin thickness of ~20 nm. Moreover, our system operates without the need for polariton condensation. Our findings highlight the promising potential of utilizing Van-der Waals materials for ultra-compact optical integrated circuits.

## Results

### Concept of an ultra-compact $WS_2$ exciton-polariton modulator

Extensive research efforts have been devoted to achieving strong coupling between photons and excitons in van der Waals semi-conductors, transition metal dichalcogenides (TMDs)[40–42]. Recent findings have revealed the natural occurrence of strong coupling in the near field of TMDs layers, resulting in the formation of guided exciton-polaritons[43]. In our previous study, we directly observed the dispersion relation of these guided polariton modes in ultrathin $WS_2$ layers (1–30 nm)[44,45]. Furthermore, we demonstrated that the $WS_2$ waveguide structure enables efficient guiding of polaritons beyond the diffraction limit[46]. Notably, the propagation loss of guided polaritons in $WS_2$ layers is significantly lower compared to surface plasmon polariton modes on metal surfaces[46]. Another significant advantage of guided exciton-polariton modes is their remarkable tunability, surpassing that of conventional optical modes. Taking these advantages of the guided polaritons, we propose their utilization for the development of ultra-compact polariton modulators.

Figure 1a illustrates the schematic diagram of the MZ modulator structure based on the guided polariton waveguide. The fabricated waveguide in the $WS_2$ modulator exhibits a width of 480 nm and a thickness of 18 nm (Fig. 1b). The calculated optical dispersion relation for the waveguide is illustrated in Fig. 1c, demonstrating the presence of polaritonic features near the exciton resonance. We also experi-mentally verified the presence of polariton modes in the waveguide, as illustrated in Supplementary Figs. 2 and 3. To resonantly excite the guided polaritons, a white light laser is focused on one end of the sample where translational symmetry is intentionally broken. It is worth mentioning that, for simplicity, a grating coupler is not utilized in our experiment. Nevertheless, integrating a grating coupler could further enhance the coupling efficiency[43,47,48]. The excited polariton flow is then subsequently split into two arms at the first beam splitter and later recombined at the second beam splitter. The recombined polariton flow along two distinct paths results in interference, given that resonantly excited polariton flow exhibits coherence. The calcu-lated polariton flow within the MZ interferometer is presented in Fig. 1d. The intensity of the interfered polaritons can be modified by manipulating the optical path length of the polariton (i.e., effective refractive index of the polariton) in one of the arms.

In our approach, we aim to alter the effective refractive index of the polariton by employing intense exciton excitation through optical pumping. We have recently discovered that the dispersion relation of the guided polariton, which corresponds to the effective refractive index, can be easily tuned by varying the laser pumping power[45]. This tuning effect arises from the redshift of the exciton resonance under high excitation pumping power. Excitons in 2D layered materials typically undergo a strong redshift under laser pumping, which can be attributed to bandgap renormalization and local heating effects[49–51]. To quantify the shift of the exciton energy as a function of excitation pumping power, we directly analyze the photoluminescence spectrum from the sample. As shown in Fig. 1e and Supplementary Fig. 4, we observed a significant exciton energy shift of up to 60 meV, resulting in corresponding changes in the polariton dispersion relations (further elaborated in the theoretical modeling section).

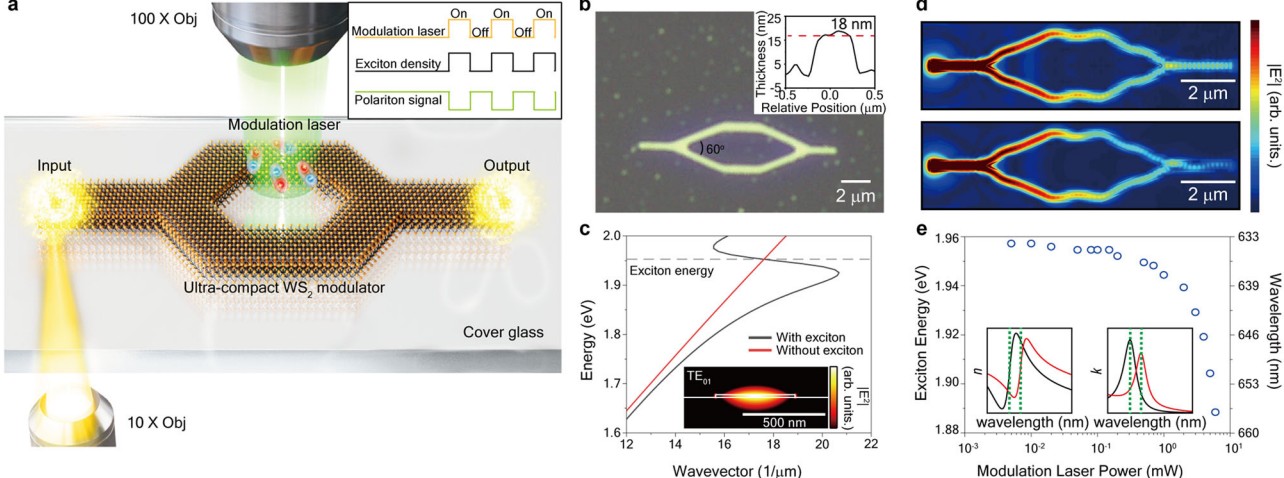

**Fig. 1 | Ultra-compact exciton-polariton Mach–Zehnder (MZ) modulator.**
**a** Schematic illustration of the ultra-compact $WS_2$ modulator. The modulation laser's on/off status induces changes in the exciton density, resulting in variations in the effective refractive index of guided exciton-polariton modes within the mod-ulator. **b** Optical microscope image of the ultra-compact $WS_2$ modulator. The detailed dimensions of the modulator are provided in Supplementary Fig. 1. (Inset) Measurement of $WS_2$ modulator's thickness using an atomic force microscope (AFM). **c** Dispersion relation of the guided mode in a $WS_2$ waveguide with and without the presence of an exciton resonance. The horizontal dashed line represents the exciton energy. (Inset) Electric field profiles of the guided polariton mode. The color bar represents $|E|^2$. **d** Simulated polariton flow in the ultra-compact $WS_2$ modulator when the modulation laser turned off (top) and on (bot-tom) at a wavelength of 705 nm. The color bars represent $|E|^2$. **e** Measured exciton energy shift as a function of modulation laser pumping power, leading to a change in the dispersion relation of the guided exciton-polariton. (Inset) The refractive index of $WS_2$ waveguide at 0 mW (black) and 5 mW (red) modulation pump-ing power.

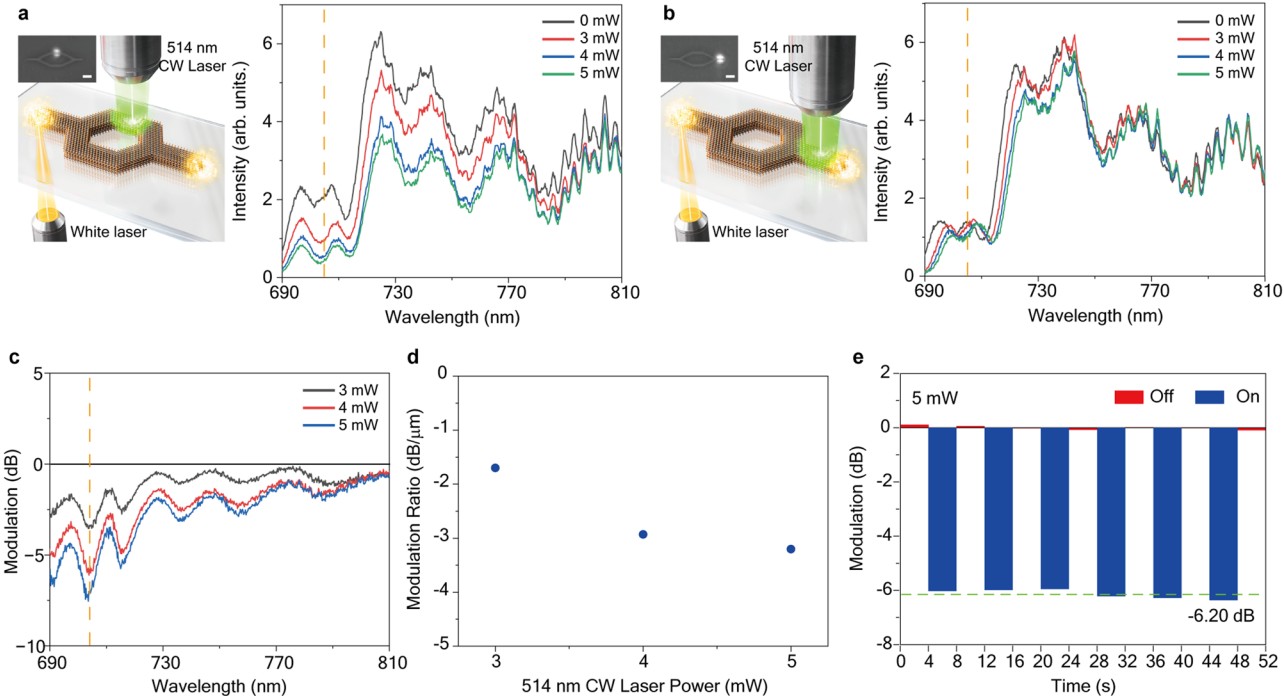

**Fig. 2 | Experimental measurement of guided exciton-polariton modulation.** Transmitted polariton spectra as a function of modulation pumping power when the modulation laser is focused on one arm of MZ modulator (**a**) and after the second beam splitter (**b**). The vertical dashed line represents the maximum modulation position at a wavelength of 705 nm. The left schematic illustration and charge-coupled device (CCD) image depict the position of the 514 nm modulation laser focusing. The scale bar is 2 μm. **c** Modulation ratio of polariton intensity calculated by $10 \log(I/I_0)$, where $I$ and $I_0$ represent the intensities of the spectrum with the modulating laser turned on and off, respectively. The vertical dashed line represents the maximum modulation position at a wavelength of 705 nm. **d** Modulation ratio per 1 μm. Modulation depending on modulating laser pumping power at the wavelength of 705 nm. **e** Repeatability of polariton modulation at the wavelength of 705 nm with continuous variation of the 'on' and 'off' status of the modulation laser.

## Experimental demonstration of guided polariton modulation

To experimentally demonstrate polariton modulation, we employed local excitation of the exciton using a modulation laser with a wavelength of 514 nm in one arm of the Mach−Zehnder interferometer, as depicted in Fig. 2a. The size of the pumping spot was approximately 2 μm², resulting in a modulation length along the waveguide of approximately 2 μm. We detected the intensity of the modulated polariton by collecting scattered light at the edge of the waveguide. It should be noted that the modulation laser also generates photoluminescence of the polariton, which contributes to an increased intensity of scattered light at the edge. However, we found this contribution to be negligible when the power of the polariton flow resonantly excited by the white light laser is sufficiently strong (see Supplementary Fig. 5).

Interestingly, we observed significant intensity modulation with different modulation pumping powers. Figure 2a illustrates the spectrum of the polariton measured at the end of the waveguide. The black graph represents the original polariton spectrum without modulation pumping. The spectral distribution of the spectrum corresponds to the polariton bandwidth, with a propagation length of approximately 14 μm, which differs from the spectrum of the white light laser (See Supplementary Fig. 6). The intensity of the polariton spectrum is notably modulated under local exciton pumping, particularly within the range of 690 nm to 810 nm (Fig. 2a). Moreover, the intensity modulation exhibits a strong dependence on the wavelength of the polariton.

The modulation laser employed in the experiment allowed for simultaneous changes in both the amplitude and phase of the polariton in one arm of the MZ interferometer. To discern the specific effects of amplitude and phase modulations, we conducted additional experiments where excitons were locally excited after the second beam splitter, affecting both polariton flows simultaneously. In this case, only amplitude modulation could alter the measured intensity.

Figure 2b displays the modulated spectrum obtained from the control experiment, which exhibited reduced intensity but in a less pronounced manner. Additionally, the modulation bandwidth was found to be limited to the shorter wavelength region. These findings from the control experiment indicate that transmission modulation in the longer wavelength region is predominantly governed by destructive interference, accompanied by an additional phase delay at the position of the modulation pumping spot. Conversely, the shorter wavelength region is primarily influenced by amplitude modulation due to the increased propagation losses under the modulation pumping, which is consistent with our theoretical modeling (see the following section and Supplementary Fig. 7).

The extinction ratio of the measured polariton modulation is illustrated in Fig. 2c, obtained by calculating $10 \log(I/I_0)$, where $I$ and $I_0$ represent the intensities of the spectrum with the modulating laser turned on and off, respectively. The modulation bandwidth covers the range from 680 nm to 780 nm. Although small oscillations of the extinction ratio are observed due to the interference pattern in the spectrum, the overall extinction ratio exhibits more pronounced modulation at a shorter wavelength. In Fig. 2d, the modulation per unit modulation length at a wavelength of 705 nm is presented as a function of modulation pumping power. Our measurements indicate modulations of −1.69 dB/μm, −2.93 dB/μm, and −3.10 dB/μm at optical pumping powers of 3 mW, 4 mW, and 5 mW, respectively. The remarkable maximum value of −3.10 dB/μm indicates highly efficient modulation, enabling the realization of an ultra-compact MZ modulator.

To assess the repeatability of the polariton modulator, we conducted a comparison of the transmitted intensity while varying the on/off state of the modulating laser at 4-s intervals. Figure 2e depicts the intensity variation at the wavelength of 705 nm under on/off pumping conditions. The intensity of the transmitted polaritons was measured

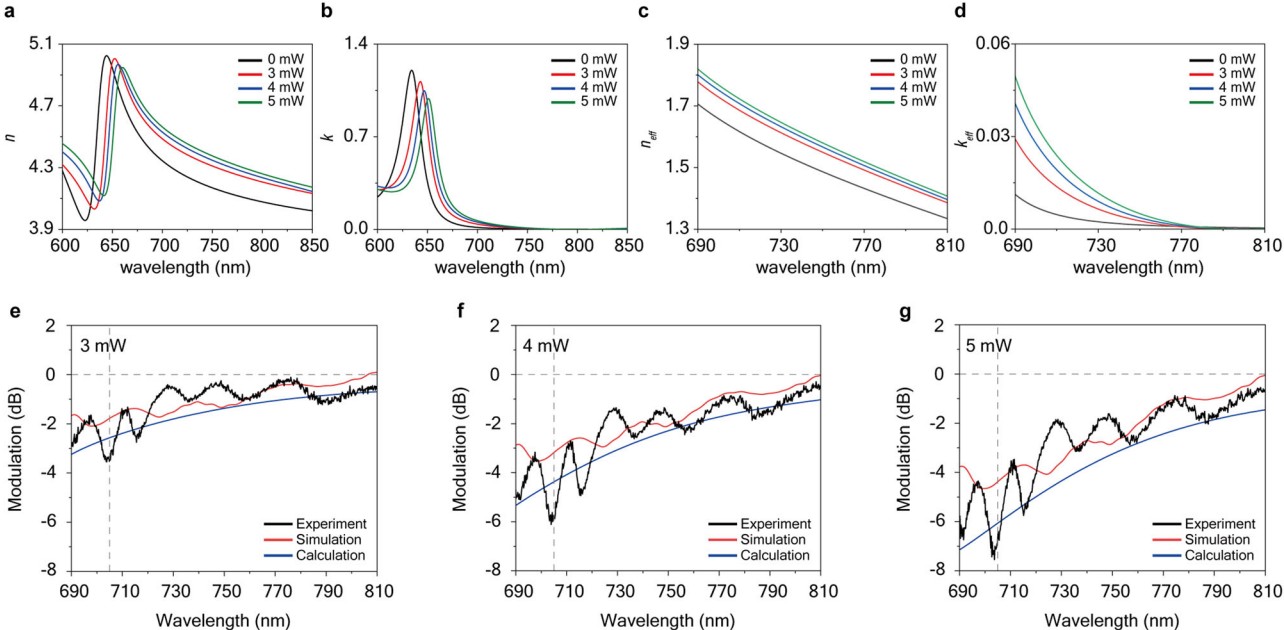

**Fig. 3 | Theoretical calculation of guided exciton-polariton MZ modulator.** Estimated real (**a**) and imaginary parts (**b**) of the refractive index of WS$_2$ multilayer with various modulation laser pumping power. Corresponding real (**c**) and imaginary parts (**d**) of the effective refractive index of guided polariton modes depending on the modulating laser pumping power. **e**–**g** Comparison of the polariton modulation obtained from experimental measurement (black lines), 3D FDTD simulation (red lines), analytical modeling with effective refractive index (blue lines) at different modulating laser pumping powers of 3 mW (**e**), 4 mW (**f**), and 5 mW (**g**), respectively.

at 4-s intervals, with an integration time of 4 s, synchronized with the laser on/off states. The results show very stable modulation with a variation of only 2.4%.

## Theoretical modeling of guided polaritons in the modulator

To understand the observed guided polariton modulation, we compared the experimental data with theoretical calculations. We first estimated the expected refractive index of the WS$_2$ layer by analyzing the measured exciton energy as a function of pumping power (Fig. 1e). The parameters of the Lorenz oscillator model were adjusted to account for the observed redshift and broadening of the exciton resonance, effectively reproducing the observed polariton modulation. As shown in Fig. 3a, b, both the real ($n$) and imaginary ($k$) parts of the refractive index of WS$_2$ exhibited redshifted behavior, indicating an increased index at a given frequency. Remarkably, this change in the refractive index closely resembled the temperature-dependent refractive index previously reported for a WS$_2$ layer[52].

Based on the estimated refractive index variations, we inferred the corresponding changes in the effective refractive index of the guided mode as a function of pumping power, as illustrated in Fig. 3c, d. Notably, both the real ($n_{eff}$) and imaginary ($k_{eff}$) parts of the effective refractive index of the guided polariton modes increased significantly with pumping power, largely due to the substantial change in the exciton energy of the material. It is noteworthy that even at the maximum modulation pumping power of 5 mW, the excitonic resonance remains robust enough to sustain the guided exciton polariton mode (refer to the dispersion relations in Supplementary Fig. 8).

We directly simulated the polariton modulations in the WS$_2$ modulator using the 3D finite difference time domain (FDTD) method. The power-dependent polariton modulation was calculated using the estimated refractive index information of the WS$_2$ material, as shown by the blue lines in Fig. 3e–g. The FDTD results exhibited good agreement with the experimental data, and the interference pattern was also visible in the modulation spectrum calculated from the FDTD simulation.

To further validate our theoretical comparison, we modeled the two-wave interference with different optical path lengths (details provided in Supplementary Information). This simple model utilized the calculated effective refractive index (both real and imaginary parts) of the guided polariton modes to calculate the intensity of the superposition of two waves. As shown by the green lines in Fig. 3e–g, the calculation results exhibited behavior similar to both the experimental data and FDTD results. These theoretical calculations confirmed that polariton modulation originated from the change in the effective refractive index of the polariton modes.

## Modulation response time of the guided polariton modulator

Finally, we analyzed the modulation speed of the WS$_2$ MZ modulator using time-resolved spectroscopy technique. As illustrated in Fig. 4a, a continuous-wave (CW) laser at a wavelength of 705 nm was coupled to the polariton flows in the MZ modulator. The selection of the 705 nm wavelength was based on both the measured modulation depth and the availability of the laser diode. The continuous flow of polaritons was modulated using a pulsed laser with an ultra-short pulse width of approximately 200 fs, enabling direct measurement of the polariton transmission's time response. The reflected signal of the modulation laser from the sample, displayed in Fig. 4b, exhibits an interval time of 12.5 ns and an instrument response time of around 250 ps.

Figure 4c, d depicts the transmitted polariton signal, exhibiting dynamic modulation in response to the pulsed laser signal. The modulation response time of 0.51 ns and 3.52 ns were estimated by fitting the falling and rising (recovering) signal with exponential functions. In addition, we also determined the switching times using the 10–90% rule to facilitate a comparison with existing literature data. Notably, our switching times, 2.75 ns for falling and 4.22 ns for rising, outpace previously reported all-optical modulators employing integrated 2D materials by three orders of magnitude[53]. Under pulsed excitation, the maximum extinction ratio was measured to be −2.11 dB. Despite expecting a higher peak power (313 W) of the pulsed laser compared to the CW laser, the maximum modulation ratio is lower in the pulsed case. We attribute this to the very short pulse width of the laser (200 fs), which may be insufficient to generate the same exciton density in the sample as in the CW case.

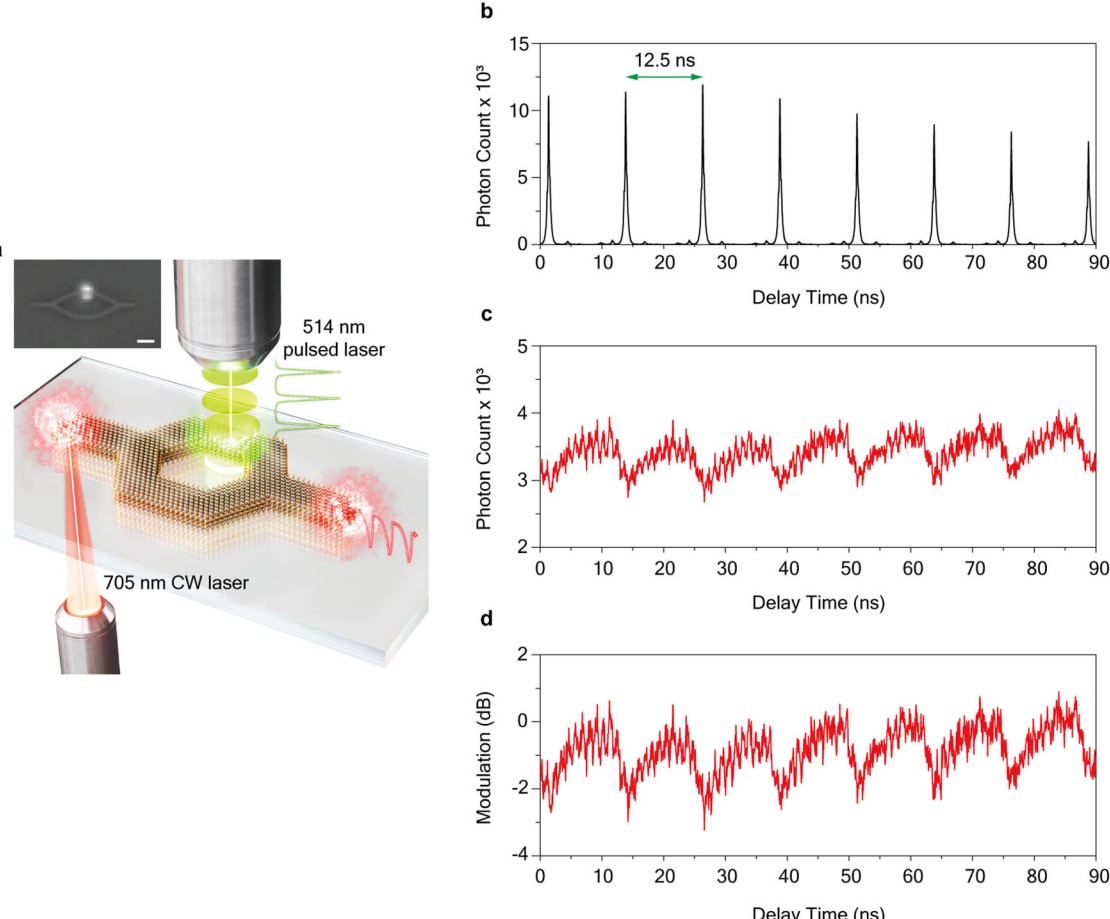

**Fig. 4 | Measurement of the modulation speed of WS$_2$ exciton-polariton modulator. a** Schematic illustration for measuring the dynamical response of polariton modulation with a 514 nm pulsed modulation laser. (Inset) CCD image showing the position of the modulation laser on one arm of the WS$_2$ modulator. The scale bar is 2 μm. **b** Time-resolved intensity of the reflected modulation laser. The interval time of the pulsed modulating laser is 12.5 ns. Measured time-resolved modulated polariton signal under the pulsed modulating laser in linear (**c**) and modulation ratio dB (**d**) scales. The exponentially fitted modulation response times are 0.51 ns and 3.52 ns for falling and rising of signal, respectively.

## Discussion

In this study, we successfully demonstrated guided exciton polariton modulation in an ultra-thin WS$_2$ MZ modulator. Remarkably, with a modulation length of only 2 μm, we achieved a significant −6.20 dB modulation ratio. Through our theoretical investigation, we have discovered that under local excitation, the change in exciton energy can substantially modify the effective refractive index of the guided polariton modes, enabling efficient polariton modulation.

The modulation principle seems similar to conventional light modulation under local laser heating, but our modulator exhibits much higher modulation efficiency. This significant difference arises from the strong exciton resonance feature, i.e., the exciton-polariton feature, of the WS$_2$ layer. The polariton with a higher exciton fraction (i.e., polariton near the exciton resonance) exhibits a more efficient modulation effect (See Supplementary Fig. 9). Even under the same excitation condition, the photonic-like mode (i.e., nearly zero exciton fraction) at a wavelength longer than 810 nm showed negligible light modulation. Additionally, the modulation laser should excite excitons to induce such an efficient modulation. When we used a modulation laser with photon energy below the exciton resonance, there was no change in polariton transmission (See Supplementary Fig. 10).

Compared to conventional dielectric waveguides like SiN waveguides, the guided polariton mode in the WS$_2$ waveguide exhibits high propagation losses due to strong excitonic absorption (See Supplementary Fig. 11). When utilizing a thick WS$_2$ waveguide with a thickness

above 100 nm, the guided mode at telecommunication wavelengths can have low optical loss[54,55]. However, since these guided modes are very far from exciton resonance, the modulation effect cannot be efficient. Thus, these propagation losses are an inherent trade-off to enable high-efficiency light modulation. The advantages of a small device footprint and shorter modulation length effectively compensate for these propagation loss issues. Notably, the footprint of our WS$_2$ MZ modulator was remarkably compact, measuring only ~30 μm², which is a few orders of magnitude smaller than conventional MZ modulators.

The results of our WS$_2$ modulator present new opportunities for ultra-compact optical interconnects and all-optical computing circuits. The potential of guided exciton polariton modulation in WS$_2$ provides a promising route toward the realization of highly integrated and efficient photonic devices, offering exciting prospects for future photonic integrated circuits based on van der Waals materials.

## Methods

### Sample fabrication

The thin WS$_2$ film with a thickness of approximately 18 nm was obtained from a mechanically exfoliated WS$_2$ bulk crystal. The WS$_2$ film was then transferred onto a glass substrate using the dry transfer method with a polydimethylsiloxane (PDMS) film. Before the nano-patterning process, a sequence of coating steps was performed to provide proper protection and support for the WS$_2$ film during the

subsequent fabrication processes. Initially, the WS$_2$ film was coated with a layer of Hexamethyldisilazane (HMD). Following that, a polymethyl methacrylate (PMMA) layer was applied to the sample surface. Additionally, an E-spacer was coated on the sample to enhance the resolution of electron-beam lithography. The designed Mach–Zehnder (MZ) modulation structure was patterned onto the sample using the electron-beam lithography technique. After the patterning step, the WS$_2$ multilayers were etched using a reactive ion etching process with a mixture of CF$_4$ (40 s.c.c.m.) and O$_2$ (10 s.c.c.m.) at 30 W, 50 mtorr, and for a duration of 1 min. Finally, the PMMA layer was removed using an acetone solution, leaving behind the WS$_2$ MZ modulator structure.

## Experimental setup

All experiments were carried out at room temperature using a custom-built microscopy setup. The polariton flow within the MZ modulator was resonantly pumped using a supercontinuum laser (NKT Photonics, Super-K laser). A high numerical aperture (NA = 1.45) oil immersion lens (Nikon, 100X magnification) was employed to tightly focus the supercontinuum laser at the edge of the MZ modulator, while the transmitted intensity was collected on the opposite side of the sample edge. For the modulation process, a continuous-wave (CW) laser with a wavelength of 514 nm was used. The modulation laser was focused on one arm of the MZ modulator using a dry lens (Olympus, 10X magnification, NA = 0.25). Collected signals were analyzed through a spectrometer (Princeton Instrument, SpectraPro HRS-300) and a CCD camera (Princeton Instrument, PIXIS 400). To analyze the response time, time-resolved spectroscopy was employed using a time-correlated single-photon counter (TCSPC, PicoHarp 300) with avalanche photodiodes (MPD, PD-020-CTF). The time bin of the TCSPC setup was set at 64 ps. In this time-resolved experiment, the polariton flow was excited using a continuous 705 nm diode laser (Thorlab, ITC4020), while the modulation was induced by a pulsed laser with a wavelength of 514 nm (Chameleon Ultra II, Chameleon Compact OPO).

## Optical simulation

The electric fields within the WS$_2$ MZ modulator structure were simulated using a 3D finite-difference time-domain (FDTD) simulator (Lumerical). The refractive index information for the WS$_2$ multilayer was obtained from ref. 56. For the simulation, the mesh size was set as follows: $x = 10$ nm, $y = 10$ nm, $z = 1$ nm. The MZ modulator had dimensions of 18 nm thickness, 480 nm width, and 13.8 μm length. To analyze the dispersion relation of the polariton modes in the MZ modulator, the finite-difference eigenmode (FDE) method was employed for simulations. Additionally, the refractive index variation under various modulation pumping power levels was estimated based on the experimental results.

## Data availability

The Source Data underlying the figures of this study are available with the paper. All raw data generated during the current study are available from the corresponding authors upon request. Source data are provided with this paper.

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

## Acknowledgements

This work was supported by the National Research Foundation of Korea (NRF-2019R1A2C2003313, NRF-2022R1A4A1034315) and National R&D Program through the National Research Foundation of Korea (NRF) funded by Ministry of Science and ICT (2021M3F3A2A03017083). We acknowledge support provided by Samsung Science and Technology Foundation (SSTF-BA1902-03) and a Korea University Grant.

## Author contributions

S.-H.G. conceptualized and supervised the study. S.W.L., J.S.L., and S.-H.G. performed all data analysis and visualization. J.S.L., S.W.L., and W.H.C. fabricated the sample structures. S.W.L. and J.S.L. conducted the optical experiments and data collection. S.W.L., J.S.L., and D.C. conducted the simulations; S.W.L. and J.S.L. wrote the original draft; and S.-H.G. revised the final manuscript.

## Competing interests

The authors declare no competing interests.
