## [Peer Review File · Nature Communications]

Ultra-compact exciton polariton modulator based on van der Waals semiconductorsREVIEWER COMMENTS

Reviewer #1 (Remarks to the Author):

The manuscript presents an intriguing study focusing on ultra-compact exciton-polariton Mach-Zehnder modulator based on WS₂ multilayers. The main contribution of the work to the field is the implication of optical modulation to a compact TMD-based MZ modulator by creating local excitation using an intense modulated laser. Such local optical excitation induces changes in the effective refractive index, which modulates the transmission of the waveguide/MZ modulator. The work presents a few significant findings, such as achieving a relatively large modulation of -6.10 dB with an ultra-short modulation of 2 μm . This is noteworthy as it indicates the modulator's effectiveness in manipulating transmitted intensity. Additionally, the ultra-compact footprint area of $\sim 30 \mu\text{m}^2$ and the exceptionally thin thickness of 18 nm are highlighted, emphasizing the potential for practical applications.

Overall, the topic is timely and relevant. While the work has several commendable aspects, there are a few concerns regarding the advancement in the field, the clarity of the authors' claims about the polaritons and its role, and the choice of spectral range for the modulator.

The current work lacks sufficient scientific novelty and clarity to be published in the prestigious Nat. Comm. journal due to the following reasons:

(i) Similar devices have been demonstrated previously and presented clear advancement and novelty of using the TMD platform for the photonics, for instance, waveguide (<https://doi.org/10.1021/acs.nanolett.3c02051>), ring resonator (<https://doi.org/10.1002/lpor.202200057>), photonic circuits (<https://doi.org/10.1002/adom.20230006>), and several others by taking their high-index and low-loss features. In terms of device design or fabrication methods, this work is based on previous literature and lacks novelty since it is neither a novel method nor a new material platform.

(ii) The main novelty of the work appears to be the implication of the optical modulation using an intense laser and achieving a relatively good modulation parameter. However, the modulation method is well-known and also expected. The fact that the device in this work is rather thin (only 18 nm) and prepared on a glass substrate (poor thermal conductor) already suggests that an intense laser can bring substantial heat locally through the cross-section of the waveguide. This leads to an effective modulation in the visible spectrum, which is not surprising.

Therefore, it would be beneficial if the authors could clarify how their work scientifically advances the TMD nanophotonics field beyond the achievements presented in the existing literatures.

Of course, I agree that the modulation of -6.10 dB in this specific platform is already good. However, it is not advancing the state-of-the-art in fast modulators; for instance, the recent work (<https://doi.org/10.21203/rs.3.rs-3151403/v1>) reports -35 dB extinction ratio.

In addition to the ordinary photonic devices, the authors claim a polariton modulation in their device in the visible region. However, this raises several concerns and requires clarification regarding the exciton-

polariton (strong-coupling) and its role in facilitating effective modulation.

(iii) The authors' use of the term "polariton" in the manuscript is concerning. It appears that the manuscript may benefit from a more precise explanation of the physical processes involved, specifically whether polaritons are indeed formed or if self-hybridization of excitons and waveguide modes better describes the phenomenon, as discussed in the existing article (<https://doi.org/10.1021/acsp Photonics.8b01194>). The authors should consider revising their terminology to avoid potential misinterpretations. I strongly encourage the authors to soften their assertion of "polariton modulation" and consider using terms such as "mode hybridization" to accurately represent the physical mechanism at play. Clarity in terminology and its role in how it improves the device's performance is essential to ensure that the scientific community understands the underlying processes.

(iv) I am concerned regarding the choice of a spectral range close to the lossy region in the visible spectrum for modulation, as opposed to targeting the Telecom range where the loss is negligibly small (<https://doi.org/10.1021/acsp Photonics.2c00433>). While the authors have demonstrated effective modulation in the visible range, the practical applicability of such devices in photonics may be limited due to high losses in the visible region. Considering the optical properties of WS₂ and other TMDC materials in the Telecom range with high refractive index and low loss, it raises questions about the strategic decision to operate in the visible range. It is beneficial to consider the potential advantages of targeting the Telecom range for on-chip photonic applications and demonstrate similar effective modulation in the Telecom.

Finally, it would benefit the authors to discuss how their work fits into the broader landscape of on-chip photonic applications and how it compares to existing materials and technologies.

In conclusion, while the manuscript presents a promising development in the field of optical modulators, it is important to address concerns related to novelty, terminology, spectral range choice, and future applications to strengthen the overall impact of this work. I believe that with revisions addressing these issues, this study could make a valuable contribution to the field of TMD nanophotonics.

Reviewer #2 (Remarks to the Author):

The authors present an interesting work on all-optical modulation of light within an integrated Mach-Zehnder interferometer based on WS₂ waveguides. The principle of work is the excitation of a guided exciton-polariton in the WS₂ waveguide and its modulation by non-local optical pumping that alters the exciton energy and thus its effective refractive index.

The manuscript is well-written and the results are very interesting for a broad audience of people interested in 2D materials and photonics in general. The observation of all-optical modulation of the exciton-polariton guided mode is reported for the first time to the best of my knowledge and for this reason this work could deserve to be published on Nat. Comm. provided that the authors address the following points:

- 1) The device is not described in detail. For example the MZI seems balanced but no description is provided. Also the angle of the y branches and associated bending radius.
- 2) Fig 1d shows the top-view of the polariton flow in the MZ interferometer with and without pump. Which is the wavelength of the propagating mode?
- 3) About fig. 1d, the mode seems not properly guided at the bend, since it seems that the mode undergoes reflections. May you clarify this aspect since it is important for the design of the MZI.
- 4) The theoretical analysis of the complex refractive index should be moved before fig. 1 since in fig. 1d it is clear is that when the pump is on, the top arm does not change only the refractive index but also the absorption. Since the phase can not be appreciated from fig. 1D, the change at the output of the MZI can be due the change of absorption (losses) in the two arms. An unbalanced MZI would have helped in observing the phase and intensity change.
- 5) Propagation losses are not addressed in the manuscript. Since this could be an advantage with respect to other technologies, the authors should address this topic.
- 6) Fig 2e shows a time varying modulation but the rise and fall time of the modulation are not reported. May you please include this information. Also fig. 2e can be improved by showing a trace of the ON/OFF collected data.
- 5) The modulation response time discussion need to be improved. From fig. 4c and 4d the rise and fall time seem to be longer than what is reported. For the estimation of the rise time a fast step signal should be used to let the signal reaching the highest extinction. For the fall time probably a larger repetition rate would be helpful since it is not clear from the figure if the highest transmission is reached or not. Probably a square wave at 50MHz could have helped. May the authors comment on this?

Reviewer #3 (Remarks to the Author):

In the manuscript titled "Ultra-compact Exciton-Polariton Modulator Based on Van Der Waals Semiconductors" by Lee et al., the authors present a study on a Mach-Zender interferometer implemented in thin WS₂ waveguides with a thickness of 18 nm. They achieve all-optical modulation of one of the interferometer arms, resulting in observed modulation of up to 6.2 dB under 6 mW continuous wave excitation. While the quality of the manuscript is commendable, there are significant issues with the discussion of the observed effects as exciton-polariton excitations, and several other points require clarification:

- It is suggested that the observed effects might be more closely related to guided modes rather than strongly-coupled exciton-polariton "dressed" states. Reference [45], which is previous work by the

authors, should be taken into account to provide context and clarify whether the observed phenomena are indeed exciton-polariton "dressed states." The authors should explain why the term exciton-polariton is used and provide evidence to support this choice.

- The use of white light excitation may not align with the authors' claim of creating exciton-polariton. Clarification is needed regarding whether the authors do not use the same excitation wavelength (514 nm) and why modulation is observed at wavelengths significantly above the typical polariton states (shown around 650 nm Extended Figure 1). Moreover, it would be helpful to show the modulation observed at wavelength below 680 nm.

- Regarding Extended Figure 1, the authors should consider showing the behavior of indirect emission in addition to the direct transition of WS₂ excitons, as it may contribute to the observed effect. Furthermore, the choice of measuring modulation at 705 nm should be justified and discussed in greater detail.

- Figure 1d differences between the two panels should be made clearer, possibly by using logarithmic maps or alternative visualization methods. Additionally, the colorbars used in the figure are not defined in either the caption or the text. There is also a typo where Figure 1c is referred to as Figure 2c (page 4, line 17).

- The manuscript suggests that the system can operate at lower pumping densities and without the need for condensation. However, the pulsed operation shown in Figure 4 appears to suggest the opposite. The authors should provide a clear explanation for this contradiction and include estimations of the fluence and power used in the pulsed experiments. Furthermore, a comparison with relevant literature regarding the figure of merit for excitation powers and observed modulation depth would strengthen the authors' claims.

- Recent literature suggests that thin WS₂ layers may contribute to cross-talk and leakage (<https://doi.org/10.1021/acs.nanolett.3c02051>). The authors should justify their choice of using thin WS₂ and consider comparing their results with thicker WS₂ waveguides to provide insights into the role of material thickness in the observed effects.

- It remains unclear why a 514 nm excitation wavelength was used in this study if the anticipated effect is the modulation of the refractive index. This choice raises questions about whether the excitation primarily targets excitons strongly coupled to the waveguide mode or an effective refractive index change.

- The authors should explain the choice and meaning of "foreign materials" at page 2, line 13.

Reviewer #1 (Remarks to the Author):

The manuscript presents an intriguing study focusing on ultra-compact exciton-polariton Mach-Zehnder modulator based on WS₂ multilayers. The main contribution of the work to the field is the implication of optical modulation to a compact TMD-based MZ modulator by creating local excitation using an intense modulated laser. Such local optical excitation induces changes in the effective refractive index, which modulates the transmission of the waveguide/MZ modulator. The work presents a few significant findings, such as achieving a relatively large modulation of -6.10 dB with an ultra-short modulation of 2 μm . This is noteworthy as it indicates the modulator's effectiveness in manipulating transmitted intensity. Additionally, the ultra-compact footprint area of $\sim 30 \mu\text{m}^2$ and the exceptionally thin thickness of 18 nm are highlighted, emphasizing the potential for practical applications.

Overall, the topic is timely and relevant. While the work has several commendable aspects, there are a few concerns regarding the advancement in the field, the clarity of the authors' claims about the polaritons and its role, and the choice of spectral range for the modulator.

The current work lacks sufficient scientific novelty and clarity to be published in the prestigious Nat. Comm. journal due to the following reasons:

(i) Similar devices have been demonstrated previously and presented clear advancement and novelty of using the TMD platform for the photonics, for instance, waveguide (<https://doi.org/10.1021/acs.nanolett.3c02051>), ring resonator (<https://doi.org/10.1002/lpor.202200057>), photonic circuits (<https://doi.org/10.1002/adom.202300006>), and several others by taking their high-index and low-loss features. In terms of device design or fabrication methods, this work is based on previous literature and lacks novelty since it is neither a novel method nor a new material platform.

Response:

We appreciate the reviewer's recommendations to include references, and we have incorporated the mentioned citations into the manuscript.

Regarding the novelty of our work, it primarily lies in the structural thickness and high nonlinearity of polariton. It is noteworthy that all the references provided by the reviewer describe devices with a thickness above 100 nm. In contrast, our study demonstrated the efficient manipulation of light at an extremely thin level, approximately 10 nm. This level of thickness has not been explored in conventional dielectric mediums.

Furthermore, our manuscript introduces active light modulation using TMD material, which, to our knowledge, has not been previously reported. The unique characteristics of exciton polaritons in TMD materials enable this efficient light modulation. We will provide detailed discussions to substantiate these claims in the following responses.

Hence, we believe our manuscript offers significant novelty compared to existing literature, both in terms of structural dimensions and the utilization of TMD materials for active light modulation.

Action taken:

We cited references the reviewer mentioned and included a discussion of the novelty of our work in the revised manuscript. (Page 14)

“When utilizing a thick WS₂ waveguide with a thickness above 100 nm, the guided mode at telecommunication wavelengths can have extremely low optical loss^{53,54}. However, since these guided modes are very far from exciton resonance, the modulation effect cannot be efficient. Thus, these propagation losses are an inherent trade-off to enable high-efficiency light modulation. The advantages of a small device footprint and shorter modulation length effectively compensate for these propagation loss issues. Notably, the footprint of our WS₂ MZ modulator was remarkably compact, measuring only ~30 μm², which is a few orders of magnitude smaller than conventional MZ modulators.”

(ii) The main novelty of the work appears to be the implication of the optical modulation using an intense laser and achieving a relatively good modulation parameter. However, the modulation method is well-known and also expected. The fact that the device in this work is rather thin (only 18 nm) and prepared on a glass substrate (poor thermal conductor) already suggests that an intense laser can bring substantial heat locally through the cross-section of the waveguide. This leads to an effective modulation in the visible spectrum, which is not surprising.

Therefore, it would be beneficial if the authors could clarify how their work scientifically advances the TMD nanophotonics field beyond the achievements presented in the existing literatures.

Of course, I agree that the modulation of -6.10 dB in this specific platform is already good. However, it is not advancing the state-of-the-art in fast modulators; for instance, the recent work (<https://doi.org/10.21203/rs.3.rs-3151403/v1>) reports -35 dB extinction ratio.

Response:

We appreciate the reviewer's insightful comments. It is crucial to emphasize that the highly efficient light modulation observed in our study cannot be solely attributed to local heating in a thin film. The novelty in our work stems from the remarkable nonlinearity exhibited by exciton polariton modes.

In the case of conventional materials, heating effects alone would not lead to efficient light modulation with a short 2 μm modulation length. This is because the refractive index change due to local heating in traditional materials is typically insufficient to produce substantial modulation. However, in our study, we harnessed the unique properties of **exciton polariton modes near the exciton resonance, where changes in the effective refractive index are significant even under the same temperature differences**. Our experimental results support this, as demonstrated in Figures 2-3. Even under the same excitation condition (i.e., the same crystal temperature), the long wavelength above ~800 nm, which is photonic-like mode with a very low fraction of exciton, exhibits negligible light modulation.

In addition, **to clarify the significance of the modulation laser wavelength, we conducted a new experiment with a 680 nm modulation laser**. Focusing the 680

nm laser on one arm of the MZI, where excitons cannot be excited, resulted in no change in polariton transmission. While sharp noisy peaks appear due to the laser tails, the overall shape of the polariton spectrum was not influenced by the 680 nm modulation laser. This result demonstrates that **exciton excitation is crucial for inducing modulation**. Firstly, the laser with exciton excitation can induce a higher crystal temperature change owing to the thermal relaxation processes of excitons. Secondly, high exciton density also causes a bandgap renormalization effect resulting in the red-shift of the exciton energy. Therefore, the resultant redshift of exciton energy under exciton pumping leads to an effective refractive index change of the guided exciton polariton.

Figure R1. Control experiment for polariton modulation with a 685 nm modulation laser.

Furthermore, it's essential to highlight the crucial role played by TMD materials in our system. **TMD materials possess an exceptionally large exciton binding energy, enabling polariton modes to persist even under local laser-induced heating.** Additionally, as discussed in the main manuscript, **TMD materials exhibit a substantial bandgap renormalization effect when subjected to local laser excitation**, contributing to the enhanced modulation effect.

In summary, our work advances the field by demonstrating the efficient modulation of light in a TMD-based ultra-thin waveguide, leveraging the unique excitonic properties of these materials in combination with local excitations using an intense laser.

Action taken:

We addressed the relevant discussion in the revised manuscript. (Page 14)

“The modulation principle seems similar to conventional light modulation under local laser heating, but our modulator exhibits much higher modulation efficiency. This significant difference arises from the strong exciton resonance feature, i.e., the exciton-polariton feature, of the WS₂ layer. The polariton with a higher exciton fraction (i.e., polariton near the exciton resonance) exhibits a more efficient modulation effect. As shown in Fig. 2, the photonic-like mode at a longer wavelength showed negligible light modulation even under the same excitation condition. Additionally, the modulation

laser should excite excitons to induce such an efficient modulation. When we used a modulation laser with photon energy below the exciton resonance, there was no change in polariton transmission (Extended Data Figure 9). ”

In addition to the ordinary photonic devices, the authors claim a polariton modulation in their device in the visible region. However, this raises several concerns and requires clarification regarding the exciton-polariton (strong-coupling) and its role in facilitating effective modulation.

Response:

We address the evidence of the polariton mode in the WS₂ waveguide in the following response. Please see below.

(iii) The authors' use of the term "polariton" in the manuscript is concerning. It appears that the manuscript may benefit from a more precise explanation of the physical processes involved, specifically whether polaritons are indeed formed or if self-hybridization of excitons and waveguide modes better describes the phenomenon, as discussed in the existing article (<https://doi.org/10.1021/acsphotonics.8b01194>). The authors should consider revising their terminology to avoid potential misinterpretations. I strongly encourage the authors to soften their assertion of "polariton modulation" and consider using terms such as "mode hybridization" to accurately represent the physical mechanism at play. Clarity in terminology and its role in how it improves the device's performance is essential to ensure that the scientific community understands the underlying processes.

Response:

We appreciate the reviewer's input and concerns regarding the terminology used in our manuscript. It is crucial to ensure precise and accurate terminology to describe the physical processes. ***As the reviewer mentioned, the polariton in our system arises from self-hybridization of excitons and waveguide modes.***

As demonstrated in Extended Fig. 8, we have shown that the optical modes in our device are exciton-polariton modes, resulting from the anti-crossing between excitons with optical waveguide modes. In our many prior publications, ***we have directly confirmed anti-crossing behavior, which is indicative of exciton-polariton modes*** [Advanced Materials 34, 2207735 (2022); Nanophotonics 12, 2563-2571 (2023); Advanced Optical Materials 11, 2300069 (2023)]

To further address these concerns and provide more substantial evidence, we have conducted new measurements to experimentally demonstrate the E-k dispersion relation of polariton modes in our system. We believe that these new results offer robust support for the presence of self-hybridized polariton modes in our Mach-Zehnder modulator.

New Experiment Results:

The dispersion relation of the guided exciton-polariton could not be directly measured using a far-field spectroscopy setup because they exist in the near-field. Therefore,

we employed two different approaches to gather direct evidence of exciton-polariton modes.

[Experimental Results 1: E-k Dispersion Measurement using Fourier Microscope]

We employed a periodic structure to fold the E-k dispersion curve of the guided exciton-polariton into the air-light cone. The period and filling factor of the 1D grating structure were set to 480 nm and 0.42, respectively. The measured E-k dispersion of the reflection of the guide mode resonance is illustrated in Figure R2.

Due to the very small width of the WS₂ waveguide (~480 nm), the signal-to-noise ratio of the data is very low. Nevertheless, the anti-crossing behavior of the polariton modes is visible near the exciton resonance. It is also important to point out that, even at the B exciton energy, anti-crossing behavior is noticeable. These results successfully indicate the presence of exciton-polariton modes in the WS₂ waveguide in the MZ modulator.

Figure R2. Measurement of the dispersion relation of guided exciton polariton with a periodic structure. (a) Angle-resolved reflection of the folded guided exciton-polariton modes with a periodic structure. (b) Cross-section of the measured reflection for angles between 21° and 33° with 3° intervals.

[Experimental Results 2: Reconstruction of E-k Dispersion from Interference Patterns]

We also attempted to measure the E-k dispersion of the guided polariton itself without folding it into the air light cone. We investigated interference peaks of guided-polariton spectrum, which is a typical method for estimating dispersion relations of a guided polariton along a waveguide [ref: PNAS 108, 10050 (2011), PRL 97, 147401 (2006)]. Because the spectral distribution of interference peaks (Fig. R3a) is strongly related to lateral momentum (q) as a function of frequency, we could directly estimate the E-k dispersion relation of guided modes from the interference peaks measured at a sample edge. In Fig. R3b, the distribution of the interference peaks (solid circles) was fitted on the calculated dispersion curves (solid lines). The interference peaks were placed in lateral momentum (q) at $q = 2\pi \left(\frac{m}{L} + \frac{1}{\lambda} \right)$. We considered the interference between the polariton path and the direct reflection of the white light laser from the glass substrate. L and m are a path difference and an integer number for the interference peaks. Thus, gradually decreasing energy spacing between adjacent interference peaks near the exciton energy (~ 1.97 eV) directly indicates the dispersion curve should be curved near the exciton resonance rather than a straight light line. These experimental data successfully support the existence of exciton polariton modes.

Figure R3. Measurement of the dispersion relation of guided exciton polariton (a) Measured spectrum of the scattered polariton at the edge of the sample, showing several interference peaks. (b) Estimated dispersion relation of the guided exciton polariton from the energy distribution of the interference peaks.

We ensure that the manuscript's terminology is accurate and appropriately describes the underlying physical mechanisms, and we hope that the reviewer finds these additional measurements and clarifications helpful.

Action taken:

We added the new experimental results supporting the presence of exciton polariton mode in Extended Data Figure 1-2.

(Page 4) "...We also experimentally verified the presence of polariton modes in the waveguide, as illustrated in Extended Data Figures 1 and 2. ..."

(iv) I am concerned regarding the choice of a spectral range close to the lossy region in the visible spectrum for modulation, as opposed to targeting the Telecom range where the loss is negligibly small (<https://doi.org/10.1021/acsp Photonics.2c00433>). While the authors have demonstrated effective modulation in the visible range, the practical applicability of such devices in photonics may be limited due to high losses in the visible region. Considering the optical properties of WS₂ and other TMDC materials in the Telecom range with high refractive index and low loss, it raises questions about the strategic decision to operate in the visible range. It is beneficial to consider the potential advantages of targeting the Telecom range for on-chip photonic applications and demonstrate similar effective modulation in the Telecom. Finally, it would benefit the authors to discuss how their work fits into the broader landscape of on-chip photonic applications and how it compares to existing materials and technologies.

Response:

We appreciate the reviewer's insightful comment regarding the choice of spectral range for modulation in our work. The reviewer correctly points out that targeting the Telecom range can significantly reduce propagation losses in transition metal dichalcogenide (TMD) materials due to low exciton absorption.

In response, we'd like to clarify our strategic decision to operate in the visible range, despite the higher losses. **The primary reason for this choice is the efficient light modulation effect achievable by utilizing exciton polariton modes, which is at a visible range.** While losses due to excitonic absorption in this range are significant, the nonlinearity of exciton-polaritons with a large excitonic fraction allows for highly effective light modulation over a short modulation length of 2 μm . **These losses are an inherent trade-off to enable high-efficiency light modulation.** It is essential to note that light modulation is negligible at longer wavelength regions in Fig. 3, where guided modes behave more like photonic modes rather than polariton modes.

The visible wavelength range, though associated with higher losses, has distinct advantages for on-chip photonic applications. For chip-to-chip communication, where shorter propagation distances are common, the compact footprint of our device is particularly beneficial for achieving high integration. Furthermore, the strong light confinement of exciton-polaritons near the exciton resonance implies that the polariton-to-electron conversion at the detector level should be more efficient compared to conventional optical modes.

We believe our work fits well within the landscape of on-chip photonic applications, particularly for scenarios where compact size, high integration, and efficient light modulation are essential. While we acknowledge the trade-off in terms of losses, the effectiveness of our modulation method and the unique properties of TMD materials make our approach a valuable addition to the field of on-chip photonic applications.

Action taken:

We added the relevant discussion in the conclusion of the revised manuscript.

“...When utilizing a thick WS₂ waveguide with a thickness above 100 nm, the guided mode at telecommunication wavelengths can have extremely low optical loss^{53,54}. However, since these guided modes are very far from exciton resonance, the modulation effect cannot be efficient. Thus, these propagation losses are an inherent trade-off to enable high-efficiency light modulation. ...”

In conclusion, while the manuscript presents a promising development in the field of optical modulators, it is important to address concerns related to novelty, terminology, spectral range choice, and future applications to strengthen the overall impact of this work. I believe that with revisions addressing these issues, this study could make a valuable contribution to the field of TMD nanophotonics.

Response:

We are sincerely grateful for the reviewer's constructive feedback and acknowledgment of the potential value of our work. The comments provided by the reviewer have been instrumental in enhancing the quality and clarity of our manuscript, and we highly value the input. Your feedback has been invaluable in improving the overall quality of the manuscript.

Reviewer #2 (Remarks to the Author):

The authors present an interesting work on all-optical modulation of light within an integrated Mach-Zendher interferometer based on WS₂ waveguides. The principle of work is the excitation of a guided exciton-polariton in the WS₂ waveguide and its modulation by non-local optical pumping that alters the exciton energy and thus its effective refractive index.

The manuscript is well-written and the results are very interesting for a broad audience of people interested in 2D materials and photonics in general. The observation of all-optical modulation of the exciton-polariton guided mode is reported for the first time to the best of my knowledge and for this reason this work could deserve to be published on Nat. Comm. provided that the authors address the following points:

Response:

We are grateful for the reviewer's acknowledgment of the interest and novelty in our work. The insightful comments provided by the reviewer have significantly enhanced the quality of the revised manuscript.

1) The device is not described in detail. For example the MZI seems balanced but no description is provided. Also the angle of the y branches and associated bending radius.

Response & Action taken:

Thanks for the valuable comment. We have added the detailed information of the MZI that we fabricated in the Extended Data Figure 3.

Extended Data Figure 3 | The detailed dimension of the fabricated polariton modulator.

2) Fig 1d shows the top-view of the polariton flow in the MZ interferometer with and without pump. Which is the wavelength of the propagating mode?

Response & Action taken:

The wavelength chosen for the simulation presented in Fig. 1d is 705 nm. We chose this wavelength because it corresponded to the maximum modulation depth observed in our experimental results. We address this information in the caption of the Figure 1. "...d, Simulated polariton flow in the ultra-compact WS₂ modulator when the modulation laser turned off (top) and on (bottom), at the wavelength of 705 nm."

3) About fig. 1d, the mode seems not properly guided at the bend, since it seems that the mode undergoes reflections. May you clarify this aspect since it is important for the design of the MZI.

Response:

We appreciate the Reviewer's valuable comment.

In this study, we employed a straightforward sample design as a proof-of-concept. As the reviewer pointed out, the design of the MZI is important to optimize the results. Particularly, the design of the beam splitter regions is crucial to reduce partial reflections. The incorporation of rounded bending may result in minimizing reflection and field radiation to the far-field.

Additionally, the width of the waveguide is also a key consideration in MZI design. We observed that a wider waveguide introduces a spatial oscillation of the polariton fields in one arm as a result of the beating between different polariton modes. It can lead to reduced spatial overlap of two polariton flows after the second beam splitter. This, in turn, diminishes the interference effect. We discovered that maintaining the width of the polariton waveguide below 450 nm is crucial to minimize the spatial beating effect.

Action taken:

We addressed this issue in the revised manuscript together with the detailed information of MZI design in Extended Data Fig. 3.

“Optimizing the design of the MZI is important for enhancing the device's efficiency. Our initial proof-of-concept design exhibited some partial reflection at the beam splitter region. Refinements such as optimizing the angle of the beam splitter or incorporating a rounded bending of the polariton waveguide would reduce unnecessary losses arising from reflection and far-field radiation. Furthermore, we observed that the width of the waveguide plays a crucial role in MZI design. A wider waveguide induces spatial oscillations of the polariton fields in one arm due to the beating between different polariton modes. This spatial oscillation results in reduced overlap of the two polariton flows after the second beam splitter, thereby diminishing the interference effect. To mitigate this, we identified that maintaining the width of the polariton waveguide below 450 nm is essential to minimize the spatial beating effect.”

4) The theoretical analysis of the complex refractive index should be moved before fig. 1 since in fig. 1d it is clear is that when the pump is on, the top arm does not change only the refractive index but also the absorption. Since the phase can not be appreciated from fig. 1D, the change at the output of the MZI can be due the change of absorption (losses) in the two arms. An unbalanced MZI would have helped in observing the phase and intensity change.

Response and Action taken:

Appreciating the reviewer's insightful comment, we have made the necessary adjustments to enhance the clarity of our manuscript. We included a schematic representation of the complex refractive index change of WS₂ upon modulation laser

excitation in Fig. 1. This addition aims to provide readers with a clearer understanding of the underlying principles of our device.

Fig. 1 | Ultra-compact exciton-polariton Mach-Zehnder modulator

Furthermore, we agree with the reviewer that differentiating between phase and amplitude modulation is important to comprehensively understand our device. Both experimental and simulation approaches were systematically employed to investigate this distinction. In the experimental setup, we intentionally varied the focusing position of the modulation laser before and after the second beam splitter, as illustrated in Fig. 2. This deliberate variation allowed us to discern amplitude and phase modulation. To further elucidate the effects of amplitude and phase modulation, we conducted analytic calculations based on the experimental data. The results of this analysis are presented in Extended Data Fig. 7, providing a clear distinction between amplitude and phase modulation. Through these combined efforts, we have concluded that phase modulation predominates in the longer wavelength region. We appreciate the reviewer's valuable comment.

5) Propagation losses are not addressed in the manuscript. Since this could be an advantage with respect to other technologies, the authors should address this topic.

Response & Action Taken:

We appreciate the reviewer's significant comment. We have addressed the propagation loss of the polariton modes in the revised manuscript (Extended Data Figure 10).

Extended Data Figure 10 | Calculated propagation loss of guided exciton polariton mode in WS₂ waveguide.

It is acknowledged that the propagation loss of the polariton modes is not advantageous compared to other technologies due to the large excitonic losses. However, these losses are an inherent trade-off necessary for enabling high-efficiency light modulation. We emphasize that, despite the losses, efficient light modulation compensates for the short propagation length, allowing for effective light manipulation in an ultra-compact optical system. We added the relevant discussion in the revised manuscript (Page 14).

“Compared to conventional dielectric waveguides like SiN waveguides, the guided polariton mode in the WS₂ waveguide exhibits high propagation losses due to strong excitonic absorption (See Extended Data Figure 10). When utilizing a thick WS₂ waveguide with a thickness above 100 nm, the guided mode at telecommunication wavelengths can have extremely low optical loss^{53,54}. However, since these guided modes are very far from exciton resonance, the modulation effect cannot be efficient. Thus, these propagation losses are an inherent trade-off to enable high-efficiency light modulation. The advantages of a small device footprint and shorter modulation length effectively compensate for these propagation loss issues. Notably, the footprint of our WS₂ MZ modulator was remarkably compact, measuring only ~30 μm², which is a few orders of magnitude smaller than conventional MZ modulators.”

6) Fig 2e shows a time varying modulation but the rise and fall time of the modulation are not reported. May you please include this information. Also fig. 2e can be improved by showing a trace of the ON/OFF collected data.

Response & Action Taken:

We appreciate the reviewer’s comment.

In Fig. 2, the white light laser, a picosecond pulsed laser, was employed to excite polariton modes at the edge of the waveguide. The CW modulation laser was turned on and off to modulate the transmitted polariton intensity. Typically, conventional spectroscopy setups employing normal CCD detectors are too slow to capture the time dynamics of semiconductor systems. Moreover, in this measurement, due to the

limitation of signal-to-noise ratio, the integration time for data collection was set at 4 seconds. Accordingly, the CW modulation laser was turned on and off with a 4-second interval, which is extremely slow compared to the response time of our polariton modulator (few ns). Figure 2e already presents the full trace of the collected on/off data. This is why we implemented a time-resolved measurement setup (with a fast APD and TCSPC) to capture the dynamics accurately. This information has been explicitly stated in the revised manuscript (Page 8)

“The intensity of the transmitted polaritons was measured at 4-second intervals, with an integration time of 4 seconds, synchronized with the laser on/off states.”

7) The modulation response time discussion need to be improved. From fig. 4c and 4d the rise and fall time seem to be longer than what is reported. For the estimation of the rise time a fast step signal should be used to let the signal reaching the highest extinction. For the fall time probably a larger repetition rate would be helpful since it is not clear from the figure if the highest transmission is reached or not. Probably a square wave at 50MHz could have helped. May the authors comment on this?

Response:

The reviewer raised a crucial point.

The rise time (referring to the fast step signal as noted by the reviewer) is determined based on the slope of the intensity change in the logarithm scale. Therefore, it does not necessitate the highest extinction. The modulation extinction (i.e., minimum transmission) is influenced by the carrier density excited by the laser. This becomes evident when considering the measurement of the time dynamics of exciton emission. Even with a \sim fs laser excitation, the rise time, significantly longer than the pulse width of the laser, can be accurately measured. Importantly, this rise time is independent of the intensity of the maximum count.

The fall time should be shorter than the repetition time of the laser. As depicted in Figure R4, following the rising and falling dynamics, a plateau of the polariton transmission is observable. This signifies that polariton transmission reverts to the highest level and persists until the next pulse.

Figure R4. Measured time-resolved modulated polariton signal under the pulsed modulating laser in modulation ratio dB scale.

The square pulse, which could be generated by current modulation for a laser diode, does not exhibit a sharp rising and falling time like the fs laser's shape. Moreover, the fs laser system in our group has a fixed repetition rate, preventing us from conducting a new experiment. Nevertheless, we emphasize again that the repetition rate of our measurement was slow enough to estimate the slow response time of the polariton.

Reviewer #3 (Remarks to the Author):

In the manuscript titled "Ultra-compact Exciton-Polariton Modulator Based on Van Der Waals Semiconductors" by Lee et al., the authors present a study on a Mach-Zender interferometer implemented in thin WS₂ waveguides with a thickness of 18 nm. They achieve all-optical modulation of one of the interferometer arms, resulting in observed modulation of up to 6.2 dB under 6 mW continuous wave excitation. While the quality of the manuscript is commendable, there are significant issues with the discussion of the observed effects as exciton-polariton excitations, and several other points require clarification:

Response:

We extend our gratitude to the reviewer for the examination and insightful comments. Each suggestion provided by the reviewer has played a pivotal role in substantially improving the overall quality of the revised manuscript.

- It is suggested that the observed effects might be more closely related to guided modes rather than strongly-coupled exciton-polariton "dressed" states. Reference [45], which is previous work by the authors, should be taken into account to provide context and clarify whether the observed phenomena are indeed exciton-polariton "dressed states." The authors should explain why the term exciton-polariton is used and provide evidence to support this choice.

Response:

We appreciate the reviewer's input and concerns regarding the terminology used in our manuscript.

To further address these concerns and provide more substantial evidence, we have conducted new measurements to experimentally demonstrate the E-k dispersion relation of polariton modes in our system. We believe that these new results offer robust support for the presence of self-hybridized polariton modes in our Mach-Zehnder modulator.

New Experiment Results:

The dispersion relation of the guided exciton-polariton could not be directly measured using a far-field spectroscopy setup because they exist in the near-field. Therefore, we employed two different approaches to gather direct evidence of exciton-polariton modes.

[Experimental Results 1: E-k Dispersion Measurement using Fourier Microscope]

We employed a period structure to fold the E-k dispersion curve of the guided exciton-polariton into the air-light cone. The period and filling factor of the 1D grating structure were set to 480 nm and 0.42, respectively. The measured E-k dispersion of the reflection of the guide mode resonance is illustrated in Figure R2.

Due to the very small width of the WS₂ waveguide (~480 nm), the signal-to-noise ratio of the data is very low. Nevertheless, the anti-crossing behavior of the polariton modes is visible near the exciton resonance. It is also important to point out that, even at the

B exciton energy, anti-crossing behavior is noticeable. These results successfully indicate the presence of exciton-polariton modes in the WS₂ waveguide in the MZ modulator.

Figure R2. Measurement of the dispersion relation of guided exciton polariton with a periodic structure. (a) Angle-resolved reflection of the folded guided exciton-polariton modes with a periodic structure. (b) Cross-section of the measured reflection for angles between 21° and 33° with 3° intervals.

[Experimental Results 2: Reconstruction of E-k Dispersion from Interference Patterns]

We also attempted to measure the E-k dispersion of the guided polariton itself without folding it into the air light cone. We investigated interference peaks of guided-polariton spectrum, which is a typical method for estimating dispersion relations of a guided polariton along a waveguide [ref: PNAS 108, 10050 (2011), PRL 97, 147401 (2006)]. Because the spectral distribution of interference peaks (Fig. R3a) is strongly related to lateral momentum (q) as a function of frequency, we could directly estimate the E-k dispersion relation of guided modes from the interference peaks measured at a sample edge. In Fig. R3b, the distribution of the interference peaks (solid circles) was fitted on the calculated dispersion curves (solid lines). The interference peaks were placed in lateral momentum (q) at $q = 2\pi \left(\frac{m}{L} + \frac{1}{\lambda} \right)$. We considered the interference between the polariton path and the direct reflection of the white light laser from the glass substrate. L and m are a path difference and an integer number for the

interference peaks. Thus, gradually decreasing energy spacing between adjacent interference peaks near the exciton energy (~ 1.97 eV) directly indicates the dispersion curve should be curved near the exciton resonance rather than a straight light line. These experimental data successfully support the existence of exciton polariton modes.

Figure R3. Measurement of the dispersion relation of guided exciton polariton (a) Measured spectrum of the scattered polariton at the edge of the sample, showing several interference peaks. (b) Estimated dispersion relation of the guided exciton polariton from the energy distribution of the interference peaks.

We ensure that the manuscript's terminology is accurate and appropriately describes the underlying physical mechanisms, and we hope that the reviewer finds these additional measurements and clarifications helpful.

Action taken:

We added the new experimental results supporting the presence of exciton polariton mode in Extended Data Figure 1-2.

(Page 4) "...We also experimentally verified the presence of polariton modes in the waveguide, as illustrated in Extended Data Figures 1 and 2. ..."

- The use of white light excitation may not align with the authors' claim of creating exciton-polariton. Clarification is needed regarding whether the authors do not use the same excitation wavelength (514 nm) and why modulation is observed at wavelengths significantly above the typical polariton states (shown around 650 nm Extended Figure 1). Moreover, it would be helpful to show the modulation observed at wavelength below 680 nm.

Response:

We appreciate the reviewer's feedback. There are primarily two distinct excitation methods for exciton-polaritons: resonant and non-resonant excitations.

- i) **Resonant Excitation:** White light excitation at the edge of the waveguide enables direct and resonant excitation of polaritons across a broad spectral range with high intensity. It is essential to highlight that resonant excitation is crucial for observing the polariton interference effect in the Mach-Zehnder Interferometer (MZI) since polaritons excited non-resonantly lack coherence.
- ii) **Non-Resonant Excitation:** Exciting excitons with a 514 nm laser can undergo relaxation to the polariton state, enabling non-resonant excitation of polaritons. The resulting polariton states have wavelength ranges of 630-680 nm (i.e., the 514 nm wavelength does not correspond to the modulated polariton wavelength). In this excitation condition, momentum conservation is not necessary for polariton excitation, allowing us to excite excitons and polaritons in the middle of the MZI. However, polaritons excited non-resonantly have negligible intensity compared to resonantly excited polariton flows. Therefore, the non-resonantly excited polaritons do not significantly influence the results; instead, the high density of excitons affects the effective refractive index for the resonantly excited polariton flow.

Extended Figure 1 illustrates the “modulation of the exciton wavelength” measured in the far-field. It's important to note that the guided exciton polariton studied in this manuscript is a near-field mode, and therefore its modulation cannot be observed in the far-field spectrum. Instead, we refer to a prior study that directly measured exciton polariton modulation using a near-field coupling method (ref: *Advanced Materials* 34, 2207735 (2022)).

For further clarification, we provide the complete modulation spectrum obtained. Wavelengths below 680 nm exhibit minimal intensity due to high propagation loss associated with exciton absorption.

Figure R5. The complete polariton modulation spectrum obtained when the modulation laser is focused on one arm of MZ modulator (a) and after the second beam splitter (b).

Action taken:

We addressed the relevant information in the revised manuscript (Pages 4-5)

“To resonantly excite the guided polaritons, a white light laser is focused on one end of the sample where translational symmetry is intentionally broken.”... “The recombined polariton flow along two distinct paths results in interference, given that resonantly excited polariton flow exhibits coherence.”

- Regarding Extended Figure 1, the authors should consider showing the behavior of indirect emission in addition to the direct transition of WS₂ excitons, as it may contribute to the observed effect. Furthermore, the choice of measuring modulation at 705 nm should be justified and discussed in greater detail.

Response & Action Taken:

Thanks for the reviewer's valuable feedback.

In Extended Figure 1, we have added a spectrum illustrating the behavior of indirect emission under the modulation laser. The transition energy of the indirect bandgap was not strongly influenced by the modulation laser power.

Extended Data Figure 4 (Previous Extended Data Figure 1) | Power-dependent spectra measured at the excitation spot. a, The normalized exciton spectrum of WS₂ multilayers under varying pumping power of the 514 nm continuous-wave laser. b, The whole measured spectrum of WS₂ multilayers. The wavelength of indirect emission (~870 nm) was not strongly influenced by the modulation laser power.

To assess the contribution of indirect bandgap emission on the polariton modulation effect, we have included the spectrum of indirect emission guided at the ends of the modulator in Extended Data Figure 6. The indirect emission, excited by the 514 nm laser, can also be guided along the WS₂ waveguide and scatter at the modulator's end edge. The intensity of the guided indirect emission increases with modulation power, which is the opposite trend concerning the polariton modulation effect. In addition, it is also crucial to emphasize that the intensity of the guided indirect emission at the edge is negligible when compared to the resonantly excited polariton flow.

The selection of the 705 nm wavelength is based on both the measured modulation depth and the availability of the laser diode. As demonstrated in our experimental results in Fig. 2, the modulation depth is maximized at 705 nm. Additionally, the white light laser used for resonant polariton excitation is pulsed, while the 514 nm modulation laser operates in continuous wave (CW). To investigate the modulation response time using a short-pulsed modulation laser, a CW laser diode with a single frequency is required for resonant polariton excitation. Therefore, we selected the 705 nm wavelength, considering the availability of a suitable laser diode for purchase. We added this information in the revised manuscript (Page 12).

- Figure 1d differences between the two panels should be made clearer, possibly by using logarithmic maps or alternative visualization methods. Additionally, the colorbars used in the figure are not defined in either the caption or the text. There is also a typo where Figure 1c is referred to as Figure 2c (page 4, line 17).

Response and Action Taken:

We appreciate the valuable comment. We have re-plotted Fig. 1d with a clearer distinction. The information of the color bar has been added, and the typo mentioned by the reviewer has been corrected.

Fig. 2 | Ultra-compact exciton-polariton Mach-Zehnder modulator

- The manuscript suggests that the system can operate at lower pumping densities and without the need for condensation. However, the pulsed operation shown in Figure 4 appears to suggest the opposite. The authors should provide a clear explanation for this contradiction and include estimations of the fluence and power used in the pulsed experiments. Furthermore, a comparison with relevant literature regarding the figure of merit for excitation powers and observed modulation depth would strengthen the authors' claims.

Response:

Thanks for the valuable feedback.

The peak power of the pulsed modulation laser is 312 W, significantly stronger than the continuous wave modulation laser. The lower modulation depth in Figure 4 compared to Figure 2 can be attributed to the short duration time (~200 fs), which might be insufficient to generate the same exciton density in the sample as in the CW case.

Regarding the comparison of previous polariton modulation devices with the condensation regime, the pumping power density difference is not coming from the modulation power but from the excitation of polariton flow. In our demonstration, we resonantly excited guided exciton polariton flow without creating a condensed polariton flow. Thus, the excitation laser power for the polariton flow does not have any threshold, while polariton condensation has a threshold pumping power

We would like to emphasize that polariton modulation without condensation also suggests the sensitivity of sample quality. Unlike polariton condensation, which demands very sophisticated control of sample quality, the fabrication of our polariton modulator is not highly sensitive.

Action Taken:

We added the information on the peak power of the pulsed modulation laser with a relevant discussion in the revised manuscript (Page 12).

“Despite expecting a higher peak power (313 W) of the pulsed laser compared to the CW laser, the maximum modulation ratio is lower in the pulsed case. We attribute this to the extremely short pulse width of the laser (200 fs), which may be insufficient to generate the same exciton density in the sample as in the CW case.”

- Recent literature suggests that thin WS₂ layers may contribute to cross-talk and leakage (<https://doi.org/10.1021/acs.nanolett.3c02051>). The authors should justify their choice of using thin WS₂ and consider comparing their results with thicker WS₂ waveguides to provide insights into the role of material thickness in the observed effects.

Response:

We appreciate the reviewer’s important comment.

As the reviewer mentioned, a relatively thick WS₂ layer has attractive advantages for photonic integrated circuits due to the high field confinement and improved cross-talk issue. Especially, when utilizing a thick WS₂ waveguide with a thickness above 100 nm, the guided mode at telecommunication wavelengths can have extremely low optical loss. However, since these guided modes are very far from exciton resonance, the modulation effect cannot be efficient. Thus, utilization of the guided exciton polariton with high propagation losses are an inherent trade-off to enable high-efficiency light modulation.

Action taken:

We added the relevant discussion in the conclusion of the revised manuscript.

“...When utilizing a thick WS_2 waveguide with a thickness above 100 nm, the guided mode at telecommunication wavelengths can have extremely low optical loss^{53,54}. However, since these guided modes are very far from exciton resonance, the modulation effect cannot be efficient. Thus, these propagation losses are an inherent trade-off to enable high-efficiency light modulation. ...”

- It remains unclear why a 514 nm excitation wavelength was used in this study if the anticipated effect is the modulation of the refractive index. This choice raises questions about whether the excitation primarily targets excitons strongly coupled to the waveguide mode or an effective refractive index change.

Response:

We appreciate the reviewer's insightful comment.

To clarify the significance of the modulation laser wavelength, we conducted a new experiment with a 680 nm modulation laser. Focusing the 680 nm laser on one arm of the MZI, where excitons cannot be excited, resulted in no change in polariton transmission. While sharp noisy peaks appear due to the laser tails, the overall shape of the polariton spectrum was not influenced by the 680 nm modulation laser. This result demonstrates that exciton excitation is crucial for inducing modulation. Firstly, the laser with exciton excitation can induce a higher crystal temperature change owing to the thermal relaxation processes of excitons. Secondly, high exciton density also causes a bandgap renormalization effect resulting in the red-shift of the exciton energy. Therefore, the resultant redshift of exciton energy under exciton pumping leads to an effective refractive index change of the guided exciton polariton.

Figure R1. Control experiment for polariton modulation with a 685 nm modulation laser.

Action taken:

We have incorporated the new experimental results into the revised manuscript (Extended Data Figure 9) and discussed their relevance in the discussion section (Page 14).

“Additionally, the modulation laser should excite excitons to induce such an efficient modulation. When we used a modulation laser with photon energy below the exciton resonance, there was no change in polariton transmission (Extended Data Figure 9).”

- The authors should explain the choice and meaning of “foreign materials” at page 2, line 13.

Response and Action Taken:

We appreciate the reviewer's feedback. In response to the comment, we have replaced the term 'foreign materials' with 'other materials' throughout the manuscript.

REVIEWER COMMENTS

Reviewer #1 (Remarks to the Author):

The authors have made extensive efforts in addressing the comments raised during the first review, and I appreciate their responsiveness to the feedback provided. However, I still have a bit of concern regarding the authors' claim about the modulation principle, particularly about the role of polaritons in achieving a higher modulation efficiency.

In the revised manuscript, the authors state, "The modulation principle seems similar to conventional light modulation under local laser heating, but our modulator exhibits much higher modulation efficiency. This significant difference arises from the strong exciton resonance feature, i.e., the exciton-polariton feature, of the WS₂ layer. The polariton with a higher exciton fraction (i.e., polariton near the exciton resonance) exhibits a more efficient modulation effect."

To fully appreciate the significance of this claim, it is crucial that the authors provide a more detailed and elaborate discussion on how exactly polariton formation contributes to the observed efficient modulation. The statement about a higher exciton fraction of polariton leading to a more efficient modulation effect needs further clarification. Specifically:

How does the formation of polaritons give rise to better modulation than conventional modulation induced by local laser heating? The authors should elaborate on the distinct features of the modulation mechanism facilitated by polaritons.

The claim that polariton with a higher exciton fraction exhibits more efficient modulation requires additional clarification.

If a higher exciton fraction is crucial for improved modulation efficiency, why is the choice of polaritons emphasized over an uncoupled exciton system?

Clarifying why polaritons are specifically advantageous for this modulation effect would enhance the scientific understanding of the authors' findings.

Therefore, the authors are encouraged to provide a more comprehensive and detailed discussion on how the formation of polaritons contributes to the observed efficient modulation in their device.

Reviewer #3 (Remarks to the Author):

I appreciate the authors' thorough response to the points raised during the initial review. The authors' commitment to enhancing clarity, methodology, and improving the overall manuscript quality is evident. Their responsiveness to feedback has significantly strengthened the manuscript, and I can recommend its acceptance.

Reviewer #1 (Remarks to the Author):

The authors have made extensive efforts in addressing the comments raised during the first review, and I appreciate their responsiveness to the feedback provided. However, I still have a bit of concern regarding the authors' claim about the modulation principle, particularly about the role of polaritons in achieving a higher modulation efficiency.

In the revised manuscript, the authors state, "The modulation principle seems similar to conventional light modulation under local laser heating, but our modulator exhibits much higher modulation efficiency. This significant difference arises from the strong exciton resonance feature, i.e., the exciton-polariton feature, of the WS₂ layer. The polariton with a higher exciton fraction (i.e., polariton near the exciton resonance) exhibits a more efficient modulation effect."

To fully appreciate the significance of this claim, it is crucial that the authors provide a more detailed and elaborate discussion on how exactly polariton formation contributes to the observed efficient modulation. The statement about a higher exciton fraction of polariton leading to a more efficient modulation effect needs further clarification. Specifically:

How does the formation of polaritons give rise to better modulation than conventional modulation induced by local laser heating? The authors should elaborate on the distinct features of the modulation mechanism facilitated by polaritons.

The claim that polariton with a higher exciton fraction exhibits more efficient modulation requires additional clarification.

If a higher exciton fraction is crucial for improved modulation efficiency, why is the choice of polaritons emphasized over an uncoupled exciton system?

Clarifying why polaritons are specifically advantageous for this modulation effect would enhance the scientific understanding of the authors' findings.

Therefore, the authors are encouraged to provide a more comprehensive and detailed discussion on how the formation of polaritons contributes to the observed efficient modulation in their device.

Response:

We appreciate the reviewer's comments. In response, we have conducted additional calculations to further elucidate the role of exciton/photon fractions in polariton modulation. In Extended Data Figure 8, we present the results of our quantum mechanical approach, which allowed us to gain deeper insights into the modulation efficiency of polariton modes as a function of exciton fraction.

The Hamiltonian of a simple two coupled oscillator system is represented by

$$\begin{pmatrix} \hbar v q & \hbar \Omega_{Rabi} / 2 \\ \hbar \Omega_{Rabi} / 2 & E_{ex} \end{pmatrix} \begin{pmatrix} \alpha \\ \beta \end{pmatrix} = E_{pol} \begin{pmatrix} \alpha \\ \beta \end{pmatrix}$$

where E_{ex} is the exciton energy, $\hbar v q$ is the photon energy, and $\hbar \Omega_{Rabi}$ is the Rabi splitting energy. v is the phase velocity of photon $v = c / n_b$ where n_b is the background refractive index without considering exciton resonance, and E_{pol}

represents upper and lower polariton energy,

$$E_{pol} = \frac{\hbar vq + E_{ex}}{2} \pm \frac{\sqrt{(\hbar\Omega_{Rabi})^2 + (\hbar vq - E_{ex})^2}}{2}$$

The Hopfield coefficients, α and β , were estimated by fitting the upper and lower polariton energy to the guided polariton dispersion curve in the WS₂ waveguide (see the figure below). Although there is a small mismatch near the exciton resonance due to the neglect of exciton loss in the Hamiltonian model, the mixing fraction of photon and exciton ($|\alpha|^2$ and $|\beta|^2$) provides valuable insights. Our calculation results directly demonstrate that the exciton fraction decreases as the polariton energy moves away from the exciton energy. Moreover, when comparing the mixing fraction results with the modulation efficiency as a function of wavelength, we observe that a higher exciton fraction results in more efficient modulation.

As mentioned in our previous response, conventional modulation induced by local laser heating corresponds to modulation results with zero exciton fraction (at a wavelength longer than 810 nm), approaching negligible modulation.

It is crucial to note that in the case of uncoupled exciton systems, as mentioned by the reviewer, a change in the exciton resonance under local laser heating does not affect the optical-guided mode because it is not coupled to the optical mode.

We believe that these additional calculations provide valuable insights into the mechanisms underlying the modulation efficiency of polariton modes in our TMD modulators.

Extended Data Figure 8 | Estimated photon and exciton fractions of guided polariton as a function of polariton energy. **a**, Calculated dispersion relation of the guided mode in a WS_2 waveguide using semi-classical model (i.e., Lorenz oscillator model) **b**, Experimentally measured modulation ratio as a function of wavelength **c**, Calculated dispersion relation of upper and lower polariton using coupled oscillator Hamiltonian model (See Supplementary Information for details). A small mismatch near the exciton resonance is due to the neglect of exciton loss in the Hamiltonian model. **d**, The mixing fraction of photon and exciton as a function of wavelength, estimated using the Hopfield coefficient of the Hamiltonian model. The results directly demonstrate that the exciton fraction decreases as the polariton energy moves away from the exciton energy. When comparing the mixing fraction results with the modulation efficiency as a function of wavelength, we observe that a higher exciton fraction results in more efficient modulation.

REVIEWERS' COMMENTS

Reviewer #1 (Remarks to the Author):

The revisions made have significantly strengthened the manuscript. Based on the thoroughness of their revisions, I am pleased to recommend its acceptance.